# Evidence for a dual-process account of over-imitation: Children imitate anti- and prosocial models equally, but prefer prosocial models once they become aware of multiple solutions to a task

Hanna Schleihauf[1,2,3]*, Stefanie Hoehl[3,4]

1 Cognitive Ethology Laboratory, German Primate Center – Leibniz Institute for Primate Research, Göttingen, Germany, 2 Social Origins Lab, Department of Psychology, University of California, Berkeley, Berkeley, California, United States of America, 3 Max Planck Institute for Human Cognitive and Brain Sciences, Leipzig, Germany, 4 Faculty of Psychology, University of Vienna, Vienna, Austria

* hanna.schleihauf@gmail.com

**Data Availability Statement:** The data and the code employed to conduct the data analysis and an example video of the study procedure is publicly

## Abstract

Children imitate actions that are perceivably unnecessary to achieve the instrumental goal of an action sequence, a behavior termed over-imitation. It is debated whether this behavior is based on the motivation to follow behavioral norms and affiliate with the model or whether it can be interpreted in terms of a behavioral heuristic to copy observed intentional actions without questioning the purpose of each action step. To resolve this question, we tested whether preschool-aged children (N = 89) over-imitate a prosocial model, a helper in a prior third-party moral transgression, but refuse to over-imitate an antisocial model, the perpetrator of the moral transgression. After first observing an inefficient way to extract a reward from a puzzle box from either a perpetrator or a helper, children over-imitated the perpetrator to the same degree as they over-imitated the helper. In a second phase, children were then presented the efficient solution by the respective other model, i.e. the helper or the perpetrator. Over-imitation rates then dropped in both conditions, but remained significantly higher than in a baseline condition only when children had observed the prosocial model demonstrate the inefficient action sequence and the perpetrator performed the efficient solution. In contrast, over-imitation dropped to baseline level when the perpetrator had modelled the inefficient actions and the prosocial model subsequently showed children the efficient solution. In line with a dual-process account of over-imitation, results speak to a strong initial tendency to imitate perceivably irrelevant actions regardless of the model. Imitation behavior is then adjusted according to social motivations after deliberate consideration of different options to attain the goal.

available at https://osf.io/nat9f/?view_only=75acceb1f39948939bacd63ac6ec3d15.

**Funding:** This work was supported by a grant from the Deutsche Forschungsgemeinschaft (DFG; grant number HO 4342/8-1, https://www.dfg.de) awarded to SH, a stipend from the Studienstiftung des deutschen Volkes awarded to HS (no grant number, https://www.studienstiftung.de), and the IMPRS NeuroCom of the Max Planck Institute for Human Cognitive and Brain Sciences, Leipzig (no award number, https://imprs-neurocom.mpg.de).

**Competing interests:** The authors have declared that no competing interests exist.

## Introduction

When encountering an unfamiliar situation, our best bet is often to observe how others around us are acting and to adjust our own behavior accordingly. Thereby, we may observe behaviors that do not make sense to us immediately, such as clinking glasses together before drinking or taking a bow when greeting another person. Learning through observing and imitating others affords humans an efficient way to learn from other people, forgoing lengthy trial-and-error learning and obviating the need for direct instruction from a teacher [1].

Intriguingly, children will even imitate actions that have no observable functional relation with the instrumental goal of an observed action sequence. This behavior, first observed in a reward-extraction task featuring a transparent puzzle box in 3- and 4-year-old children [2] was later also documented in adults [3, 4], and has been dubbed *over-imitation* [5]. Since its discovery by Horner and Whiten [2], a range of studies have confirmed the robustness of over-imitation in human children and adults [6], across different cultures [7, 8]. At the same time, no evidence for over-imitation was found in non-human primates [2, 9, 10]. Strikingly, chimpanzees, bonobos and orangutans will directly pursue the goal of a demonstrated action sequence and drop any action they identify as irrelevant to reaching said goal while humans will seemingly waste time and effort on obviously functionally irrelevant demonstrated actions. Why is that?

Several theoretical accounts have been put forward to make sense of this peculiar human behavior. The first accounts, based on initial findings with child participants, assumed that children do not even put into question the relevance of intentional actions they observe in others. According to the "copy first–refine second" account [11], it makes sense for humans to initially "blanket copy" a range of observed actions, given that intentional actions do tend to have a purpose, even though their functional role might not always be obvious. Accidentally transmitted irrelevant actions can then be weeded out later on. Similarly, the "automatic causal encoding" account proposed that children may automatically encode observed intentional actions as causally relevant and reproduce them without much deliberation, even under time pressure [5].

Though intuitively plausible, these accounts have been criticized for neglecting the social influences on children's imitation behavior. For instance, children will reproduce irrelevant actions even after observing a more efficient solution when the inefficient model stays in the room with them, which led to the suggestion that children may over-imitate in order to affiliate with others [12]. In addition, children may interpret functionally irrelevant actions in terms of behavioral norms and reproduce them in order to adhere to these norms [13–15]. A range of studies has since provided evidence that social motivations at least have an influence on children's over-imitation [16, 17]. Over and Carpenter therefore argued that whether children imitate faithfully or not depends on their motivations in the given situation, which can be either focused on an instrumental learning goal (leading to the omission of irrelevant actions) or social goals (leading to over-imitation), or a combination of both [18]. Relatedly, it was proposed that cultural learning is driven by two distinct cognitive stances (i.e., interpretive modes): an instrumental stance (i.e., seeking out a rationale for actions based on physical causation) and a ritual stance (i.e., seeking out a rationale for actions based on cultural convention), with the latter leading to higher imitation fidelity [19–23]. A ritual stance is triggered by different cues, such as causal opacity (i.e., missing perceptibility of an action's causal purpose; [24]), start- and end- state equivalence [17], normative language [25], the presence of another person [12], or by ritual-like action characteristics such as repetition or redundancy [23, 26].

Though compelling, these theoretical accounts are not consistent with some empirical findings. Specifically, the type of demonstrated actions plays a role in over-imitation studies.

Actions performed directly on the reward container induce the highest imitation rates whereas actions performed without direct contact are imitated rarely [5, 6, 27]. It is not obvious why the social motivation to imitate faithfully should vary depending on whether the action is performed on the reward container or not.

Trying to explain these inconsistencies, Schleihauf and Hoehl introduced a dual-process model of over-imitation [28]. They argue that certain action types or task contexts (e.g., time pressure, cognitive demand) trigger blanket copying, in the vein of a Type-1-processing heuristic, which is quick and requires little cognitive effort. Specifically, when observing pseudo-instrumental actions (actions that are very similar to everyday actions that lead to an effect) performed on a reward container, such as pushing a nonfunctional lever or button, children may employ fast and cognitive efficient Type-1 processing, leading to a blanket copying of the pseudo-instrumental actions without even questioning the purpose of the action. In contrast, a physically disconnected action (that is clearly recognized as having no causal effect) will trigger deliberate consideration about its purpose. Then, the decision to copy or omit the unnecessary action is often based on the social motivations in the concrete situation. For example, when actions are perceived to be normative or ritualistic, they are more likely to be copied than actions that do not seem to serve such functions. Similarly, realizing that there is more than one solution to solve a problem will lead to deliberate consideration and may result in over-imitation when social motivations are high, or the omission of irrelevant actions, when instrumental learning goals are more prevalent.

In the current study we put the dual-process model to a critical test. If it is correct that children will blanket copy observed pseudo-instrumental actions without consideration of social motivations, they should even over-imitate a highly antisocial and therefore unlikable model, at least until they encounter a more efficient solution to attain the reward and having the chance to reflect on their options. Do children over-imitate a model that does not induce a motivation to affiliate? Would they even over-imitate a moral transgressor?

Previous research demonstrated that preschoolers detect third-party moral transgressions and show concern for the victim [29]. Already toddlers direct sympathetic concern and prosocial helping behavior towards the victim of a moral transgression, even in the absence of an overt emotional response of the victim [30]. What is more, 3-year-olds selectively avoid helping an adult who had perpetrated a third-party moral transgression, even taking into account whether the harmful action had been intentional or not [31]. Children's sensitivity to moral transgressions seems to be even more deeply rooted in human ontogeny. Several studies showed that young infants prefer a prosocial agent over a neutral agent and a neutral agent over an antisocial agent [32–34]. This body of research suggests that children should be less motivated to "be like" a moral transgressor or a build an affiliation with them. They should consequently not be eager to match their actions to those of antisocial others.

Only few, but important findings on this notion can be found in the literature: When choosing what to eat 16-month-olds took into account the preferences of a prosocial agent or a novel neutral agent, but they did not take into account information from an antisocial agent [35]. The study leaves open whether a reduced motivation for affiliation with and "being like" the antisocial agent drove this effect or whether infants put less trust into receiving reliable information from an antisocial other. Nevertheless, it is striking that at a remarkably young age, humans' behavior choices are influenced by moral evaluations.

Wilks and colleagues tested whether observing antisocial behaviors from in-group members in a minimal group setting affects children's inclination to imitate relevant and irrelevant actions from an in-group member [36]. They found that after observing two equally inefficient ways to complete a task, children aged between 4 and 8 years, would stick with imitating the actions demonstrated by their in-group over the out-group, irrespective of their group's

previously demonstrated prosocial or antisocial behavior tendencies. Imitation of relevant and irrelevant actions was not reported separately, but given that very obviously functionally irrelevant no-contact actions were used in the over-imitation tasks and that children observed two different solutions for each task, we can safely assume that children's behavior was based on deliberate considerations whom to imitate, not Type-1 blanket copying. Thus, children likely experienced conflicting social motivations and most of them (though not all) chose to stick with their in-group despite the moral transgressions displayed by several of their group members. In a related study, 4-to-8-year-old-children in Borroloola and Brisbane could choose to imitate or not to imitate clearly non-functional actions demonstrated by a prosocial model in one condition or an antisocial model in another condition [37]. While children's culture seemed to influence their imitative tendencies, with Borroloola children imitating with a higher fidelity than Brisbane children, the model's moral valence did not affect their imitation rates. The authors argued that children might be more discerning in their imitation preferences when presented with clearly distinguishable irrelevant and goal-relevant actions, copying both irrelevant and relevant actions from a prosocial model, while only copying relevant actions form an antisocial model.

In the current experiment we tested this prediction against the prediction of the dual-process model [28]. We included 5-year-old children because over-imitation has been documented reliably in this age group in previous research [6] and we know that children by this age are already highly sensitive to moral transgressions in third-party contexts [29, 31, 38]. We used a two-phase paradigm [39], in which children first observed either a prosocial or an antisocial model extract a reward from a transparent puzzle box and then were allowed to extract a reward themselves. Importantly, in the first phase the model demonstrated an inefficient solution, including both relevant and irrelevant action steps. The irrelevant action steps were further divided into pseudo-instrumental actions involving contact with the reward container and no-contact actions. Based on the dual-process model of over-imitation, we hypothesized that children would engage in blanket copying in this phase and over-imitate especially the pseudo-instrumental actions to reach the instrumental goal of extracting the reward. We predicted no difference in the over-imitation rates between children who observed a prosocial vs. anti-social model in this phase as children's behavior should be guided by Type-1 processes, foregoing a deliberate evaluation of the necessity of action steps or the morality or likability of the model.

In the second phase of the experiment, we disclosed the efficient solution to the puzzle box to the children: The respective other model now demonstrated the reward extraction directly, without any irrelevant action steps. This resulted in two between-participants conditions: antisocial model first (inefficient), then prosocial model (efficient) and vice versa. In this second phase of the experiment, due to now being aware of the efficient solution, we expected children to engage in Type-2-like processing and to deliberately consider whether to over-imitate or not. We now predicted that social motivations should come into play: Children should be less motivated to continue performing irrelevant actions that were demonstrated by an antisocial model in phase 1. Furthermore, they should be motivated to align their behavior with the prosocial efficient strategy model, resulting in a drop of their imitation rates to baseline level in this condition in the second phase of the experiment. Those children who were demonstrated the inefficient solution by the prosocial model, in contrast, may still be more inclined to follow the prosocial model's lead, and at the same time, to contrast their behavior to the antisocial efficient model. Thus, they may deliberately engage in over-imitation to a greater degree than children in the other group, despite being now aware of the most efficient way to extract the reward.

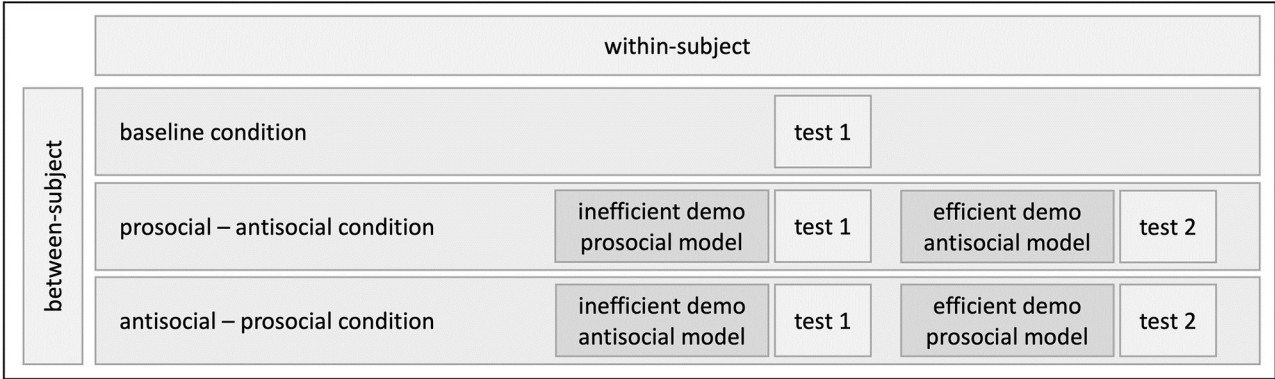

**Fig 1. Overview of baseline and experimental conditions.**

## Methods

### Design

We used a mixed design, with between-subject factors (conditions: prosocial-antisocial, antisocial-prosocial) as well as within-subject factors (test 1 and 2; see Fig 1). Children were assigned to one of two conditions: a) the *prosocial-antisocial* condition, in which the prosocial model demonstrated the inefficient strategy first, and the antisocial model demonstrated the efficient strategy second, and b) the *antisocial-prosocial* condition, in which the antisocial model demonstrated the inefficient strategy first and the prosocial model demonstrated the efficient strategy second. Additionally, we conducted a *baseline* condition in which children did not get any prior demonstration and participated only in one test trial. This condition informed us about how many irrelevant actions children would show spontaneously on our apparatus without prior social demonstration.

### Participants

The study was conducted in a medium-sized German university town. The final sample contained data of $N = 89$ children aged from 4.89 to 6.45 years (M = 5.60 years), 28 (14 girls) children in the *prosocial-antisocial condition*, 33 children (14 girls) in the *antisocial-prosocial condition* and 28 children in the *baseline* condition (15 girls). Data from an additional 3 children were not included in the analysis due to experimenter error. We calculated a power analysis, which led to the conclusion that we need a minimal sample size of 24 children per condition. This was based on baseline-comparison effect sizes in previous work on overimitation [39]: Expected effect sizes d ≥ 0.6, p < .05 (two-tailed), 1-β = 0.8. Based on our experience with this paradigm, we estimated that 5–6 additional children would have to be tested but excluded per condition due to refusal to cooperate or experimenter error.

Participants were recruited from local preschools in a mid-sized German city with around 600,000 inhabitants. Although we did not individually assess children's socio-economic backgrounds, child participants in studies conducted in urban Germany are typically from mixed to high socio-economic backgrounds. The majority of the children participating were White and spoke German as their native language. Parents in Germany and other Western, industrialized societies typically encourage their children's psychological autonomy early on [40], so it can be assumed that children knew that it was their own choice how to behave (this was also emphasized by the experimenters). Children in Germany receive high levels of direct, child-

centered pedagogy and are used to dyadic settings such as the procedure of the current study. Previous studies reported over-imitation among children from this population [16, 41, 42].

The testing took place at children's local day-care centers with the written informed consent of their parents and verbal consent of the children. Procedures were approved by the local ethics committee of the Max Planck Institute for Cognitive and Brain Science, Leipzig, Germany.

## Material

Children interacted with a transparent puzzle box containing several golden marbles. A small magnet was attached to each marble, so that it could be extracted from the box using a magnetic rod. A black wooden lever was attached to the top of the box and a black button was attached to the side of the box (see Fig 2). It was possible—due to the transparency of the box —to notice that some of the demonstrated actions were not causally necessary to extract the marbles.

## Efficient / inefficient strategy

Children observed two types of strategies of how to extract a marble from the puzzle box, one inefficient strategy and one efficient strategy. The inefficient strategy included 4 causally irrelevant actions followed by one relevant action. Two of the irrelevant actions were pseudo-instrumental actions that involved physical contact with the puzzle box, the other two were disconnected actions that did not involve contact with the box. The efficient strategy only included the relevant action (see Fig 2).

## Procedure

**Warm-up phase.** Children were tested in quiet rooms in their kindergartens with three experimenters, a prosocial, an antisocial and a neutral experimenter. The procedure took approximately 25 minutes. At the beginning of each testing, all three experimenters and the child played a warm-up game together during which the child and the experimenter played a game of blowing cotton balls into a goal to win golden marbles. During this phase the child got familiarized with the experimenters and could learn that golden marbles could be exchanged for stickers.

**Prosocial / antisocial manipulation.** We adapted the procedure of Vaish et al. [30]. The child was told that all four of them (the child and the three experimenters) would now each color a picture of their choice. After everybody had chosen a picture, the child and the

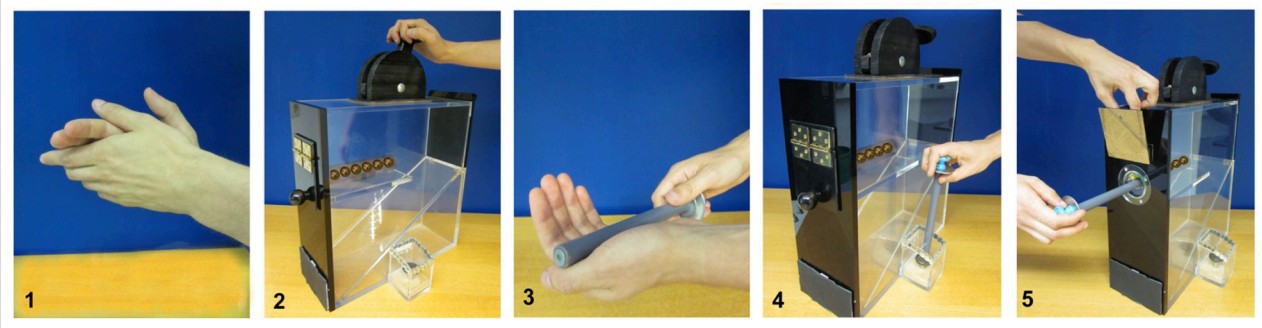

**Fig 2. The demonstration of the inefficient strategy included (1) clapping, (2) pushing the lever back and forth, (3) tapping the rod into the plam of the hand three times, (4) using the rod to push the button on the side of the box and (5) inserting the magnetic rod to extract a marble.** The demonstration of the efficient strategy only included (5) the insertion of the rod.

experimenters sat down at a table and started painting. The child sat in between the prosocial and the antisocial experimenter, and opposite of the neutral experimenter, so that they could clearly observe the interaction between all experimenters. After approximately 5 minutes of coloring, the neutral experimenter stopped, showed her picture to the rest of the group, and said: "Look what I painted! Isn't that pretty? It makes me so happy that I painted such a pretty picture." Following this, the antisocial experimenter took the picture of the neutral experimenter, said "I am going to tear up your picture!", ripped the picture into small pieces, and through it into a trash can standing right next to them. Reacting to this transgression, the neutral experimenter expressed sadness. Then, the prosocial experimenter intervened by turning to the neutral experimenter and handling her a new picture: "Poor you! Now you have no more picture. Here is a new one, so that you don't need to be sad anymore." Following this, the neutral experimenter announced: "Ok, I am going next door and continue painting there." The neutral experimenter left and did not return until the end of the testing session.

**Over-imitation.** The over-imitation part of the experiment was structured in two phases. In the first phase, one of the experimenters demonstrated the inefficient strategy, then it was the child's turn (test 1). In the second phase, the other experimenter demonstrated the efficient strategy, and then it was the child's turn again (test 2). Whether the prosocial or the antisocial experimenter demonstrated either strategy was manipulated between conditions (*prosocial-antisocial* vs. *antisocial-prosocial*).

In the *prosocial-antisocial condition* the prosocial experimenter started by introducing the box: "Next it is about this special box." Then the antisocial experimenter continued: "In this box there are more golden marbles which can be exchanged for stickers. I now need to write something down. I'll be back in a few minutes". Then the antisocial experimenter turned away and sat down by a table turning her back to the scenery. The prosocial experimenter announced "Ok, let's start then. It's my turn first" and extracted a marble using the inefficient strategy involving irrelevant actions. In the following, she explained that it would now be the child's turn ("It's your turn next.") to retrieve a marble. She explicitly stated that the child could accomplish this goal however they liked ("You can do this however you like."). Before the child started, both experimenters said that they had to go out of the room for a moment, so that the child remained alone during the test trial. As soon as the child had successfully retrieved a marble, the experimenters returned. The prosocial experimenter helped the child to choose a sticker from a sticker box in exchange for the marble, and then stated that she had to write something down and sat down by a nearby table facing away from the others. Next, the antisocial experimenter explained that it was her turn. She extracted a marble using the efficient strategy, with no causally irrelevant actions. In the following she explained that it was the child's turn again and that the child could do it however she liked, while both experimenters left the room. Once the child had extracted a marble, the experimenters returned, and the antisocial experimenter helped the child to exchange the marble for a sticker. In the *antisocial-prosocial condition*, the role of the experimenters was reversed. Otherwise the procedure was the same.

Children in the *baseline condition* did not experience the prosocial-antisocial manipulation, never saw a demonstration and participated in one single trial. As in the experimental conditions, one experimenter introduced the box and placed it in front of the child together with the magnetic rod. Then the child was told that they could extract a marble from the box however they liked. Next, the experimenter left the room and waited either until the child extracted a marble or until two minutes had passed. This condition was included in the analysis to check if performance levels of irrelevant actions exceeded spontaneous performance levels of these actions due to curiosity or exploration.

## Coding and reliability

Since we were interested in whether or not children adopt irrelevant actions, we applied a binary coding (0 = action not performed, 1 = action performed) for each action (see [43] for a systematic comparison of different over-imitation coding schemes). The children received a score of 1 for each action if they performed it at least once. The number of times they performed a single action and the order in which the irrelevant actions were performed were not considered in the scoring. Thus, clapping was coded as performed when they clapped at least once. Pushing the lever was coded as performed as soon as the lever was pushed in one direction. Tapping was coded when they tapped the rod into their palm at least once. Pushing the button was coded as performed as soon as the button was pushed down (no matter if this action was performed with the rod or with the finger).

These target behaviors were coded from video recordings of children's behavior during each imitation test trial. The primary coder was a research assistant who was blind to the hypotheses of the study. A second coder, also blind to the hypotheses, rated 25% of the videos. The inter-rater reliability (Cohen's Kappa) was excellent (reliability of 1.0), suggesting reliable assessments of over-imitation scores.

## Statistical analysis

To test our hypotheses, we applied one main logistic Generalized Linear Mixed Model (GLMMs) [44] fitted via maximum likelihood using the statistical program R (version 3.4.3) [45] together with the function 'glmer' of the package 'lme4' [46]. Furthermore, we conducted four Generalized Liner Models (GLMs) for baseline comparisons using the function 'glm' of the same package. In the main GLMM we focused on the effect of the manipulations in the experimental conditions. In the baseline comparisons we compared children's tendency to over-imitate in the experimental conditions to children's tendency to perform irrelevant actions in the baseline condition. Effect sizes were extracted with the functions 'r.squaredGLMM' and 'r.squaredLR' of the package 'MuMIn'. A detailed description of this analysis including all assumption checks can be found in the (S1 Text).

**Main GLMM.** We were mainly interested in whether children's over-imitation differs depending on condition and test phase. Thus, statistically we looked at whether the interaction between condition and test phase (1, 2) had a significant effect on children's tendency to over-imitate. Since previous research has shown that not all types of actions are over-imitated to the same extent, we included the three-way-interaction between condition, test phase and action type into our main model. Prior research also revealed gender effects showing that boys over-imitate more than girls [16, 47]. This is why we also checked whether children's gender predicted their tendency to over-imitate. To account for repeated measurements, we included the random effect for individual's identity and the random slopes for test phase and action type within individual identity in the model equation of the GLMM. Taken together, this led to the following model equation:

$$\text{over} - \text{imitation}_{(yes/no)} \sim (\text{condition}_{(prosocial-antisocial/antisocial-prosocial)} *$$
$$\text{phase}_{(first-inefficient\ demo/second-efficient\ demo)} * \text{action type}_{(contact/noncontact)}) +$$
$$\text{child's gender}_{(male/female)} +$$
$$(1 + \text{phase}_{manually\ dummy\ coded\ and\ centered} +$$
$$\text{action type}_{manually\ dummy\ coded\ and\ centered} \| \text{individual identity})$$

We first tested the overall effect of all predictors and the included interactions. Therefore, we compared the full model's deviance with that of a null model comprising only random effects to examine whether the inclusion of the test predictors provided a better fit to the data than participant identity alone. To determine the effects of each predictor we further compared the full model with the corresponding reduced models that lacked the predictor of interest. Thereby, to test the effect of the lower order interactions and the main effects we dropped the higher order terms from the model before we compared it with the respective reduced models. Since our main interest was whether children's over-imitation differs depending on condition and test phase, we performed post-hoc pairwise comparisons of both phases in each condition using the package 'emmeans'.

**Baseline comparisons.** We further tested whether children's probability to over-imitate in each phase of each condition exceeded the probability of children in the baseline condition to perform the irrelevant target actions. Therefore, we performed four additional GLMs to compare the probability to over-imitate for each test phase (1, 2) as a function of experimental conditions (*prosocial-antisocial*, *antisocial-prosocial*) with the baseline condition (*no demonstration*). Each GLM included the baseline condition as well as one test trial of each condition. To avoid inflating the Type I error rate, a Bonferroni correction was applied for four baseline comparisons (adjusted $\alpha' = 0.0125$). For the baseline comparisons we used the following model equation:

$$\text{over} - \text{imitation}_{(\text{yes/no})} \sim \text{condition}_{(\text{one phase of the experimental conditions/baseline})}$$

The data, the code for the analysis, and an example video of the procedure (visible experimenters consented to the video being published) are made available online on https://osf.io/nat9f/?view_only=75acceb1f39948939bacd63ac6ec3d15

## Results

### Main GLMM

The main GLMM provided a significantly better fit to the data than the null model ($\chi^2 = 77.984$, $df = 8$, $p < .0001$, $R^2$-like effect size: 0.63). The comparison of the full model with the respective reduced models revealed that neither the three-way interaction *condition*phase*action type* ($\chi^2 = 0.015$, $df = 1$, $p = .901$, Nagelkerke's $R^2 < .001$), nor the two-way interaction condition*phase reached statistical significance ($\chi^2 = 0.426$, df = 1, $p = .514$, Nagelkerke's $R^2 = .001$). However, the main effect for the factor *phase* was significant ($\chi^2 = 25.686$, df = 1, p < .001, Nagelkerke's $R^2 = .07$). The model predicted a higher probability for children to over-imitate in the first phase (43.6% probability to over-imitate), when they observed the demonstration of the inefficient strategy, and a lower probability in the second phase (17.7% probability to over-imitate), when they observed the efficient strategy. The factor *action type* also had a significant effect on children's tendency to over-imitate ($\chi^2 = 44.167$, df = 1, p < .001, Nagelkerke's $R^2 = .13$). The model predicted that children are more likely to over-imitate contact actions (55.8% probability to over-imitate) and less likely to over-imitate noncontact actions (9.6% probability to over-imitate). The main effect of *condition* was not significant ($\chi^2 = 2.483$, $df = 1$, $p = .115$, Nagelkerke's R2 = .007). Children's gender had no influence on their over-imitation ($\chi^2 = 1.047$, df = 1, p = .306, Nagelkerke's R2 = .003). The data and the model lines for the point estimates of GLMM are depicted in Fig 3. Additional descriptive information of the data can be found in Table 1. Post-hoc pairwise comparisons confirmed the results of the GLMM. When comparing children's over-imitation tendencies after the irrelevant demonstration (phase 1), we did not find a difference between conditions (z-ratio = 1.792, *p* = .233). Children over-imitated at high rates, no matter whether a prosocial or antisocial model performed

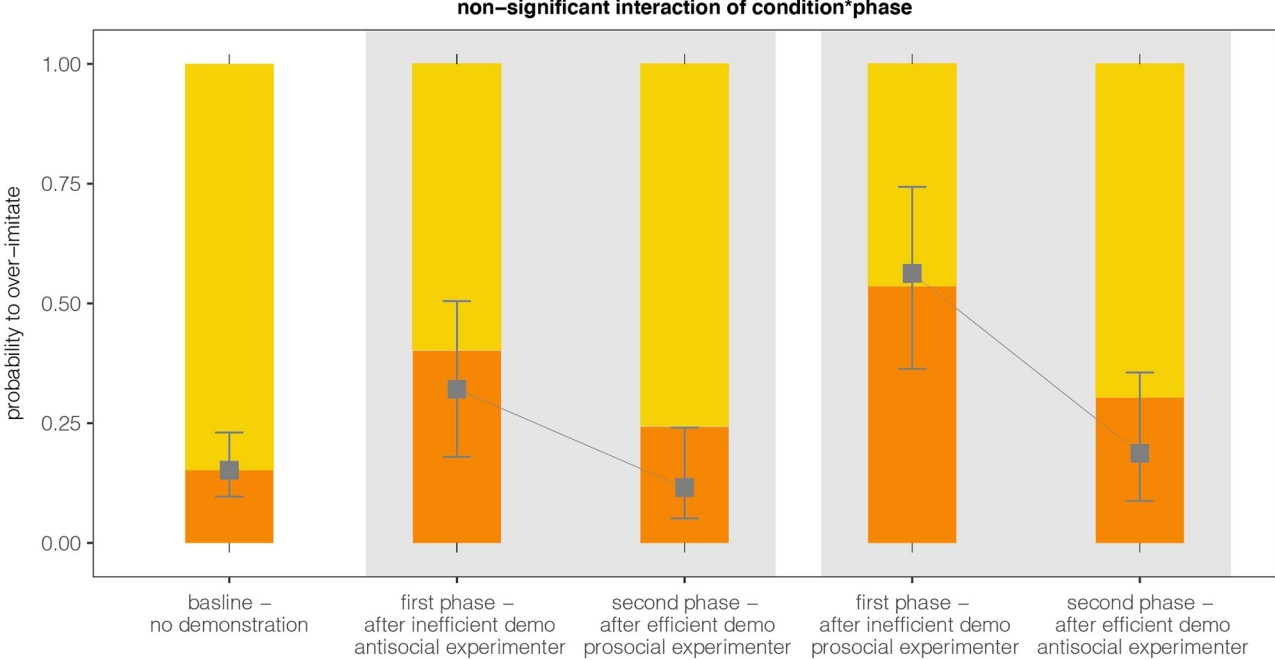

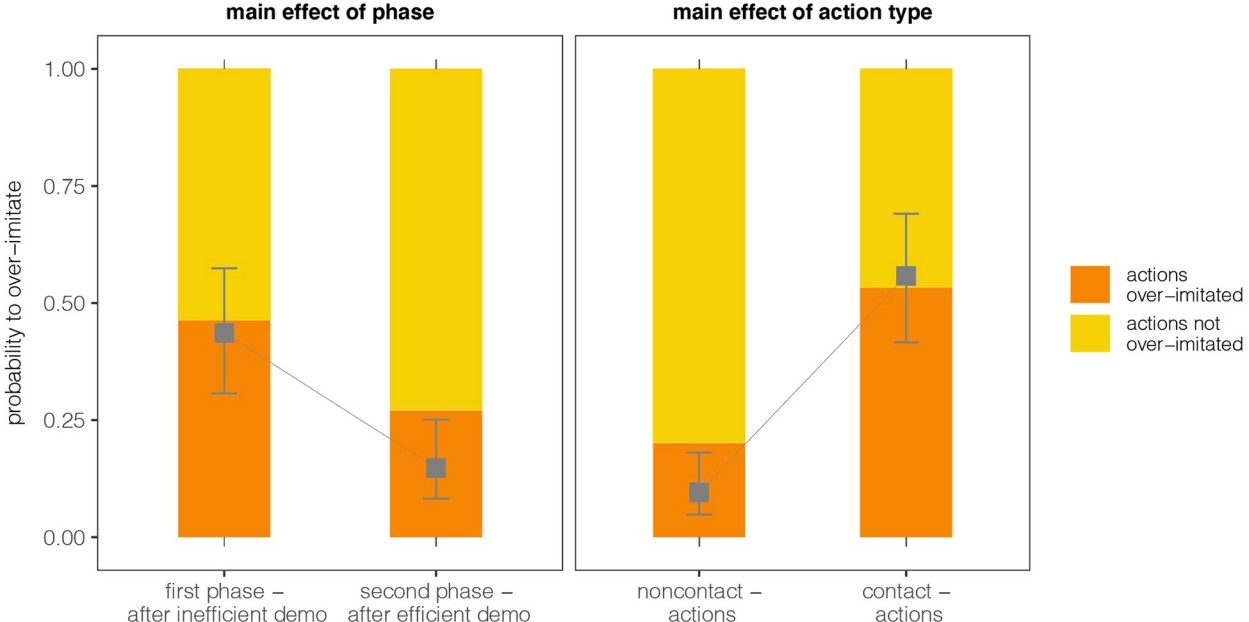

**Fig 3. The barplots show the percentages of imitated (orange) and not imitated actions (yellow).** The dots represent the point estimates of the GLMM (centered for all other predictors) with the corresponding 95% confidence intervals that were calculated with the package emmeans. Upper figure: Probability of children to over-imitate as a function of condition and test phase. Asterisks depict the results of the post-hoc pairwise comparison (*p*-value was mvt adjusted for four tests), and for baseline comparisons (*p*-value was Bonferroni adjusted for four tests). Lower left figure: Probability of children to over-imitate as a function of the significant effect of phase. Lower right figure: Probability of children to over-imitate as a function of the significant effect of action type.

the demonstration. Also, when comparing children's over-imitation tendencies after the relevant demonstration, we did not find a condition difference (z-ratio = 1.118, *p* = .633). When comparing children's over-imitation between phase 1 and phase 2 within a condition, we found a significant drop in over-imitation in both conditions (prosocial-antisocial/phase 1 –

**Table 1. Descriptive information: Number of children who re-enacted each of the four irrelevant actions, mean oi score and standard deviation for each test trial of each condition.**

| Condition | Frequency of each non-functional action performed | | | | Frequencies of sum of imitated actions | | | | | Mean sum score (SD) |
|---|---|---|---|---|---|---|---|---|---|---|
| | Clapping | Pushing lever | Tapping stick | Pushing button | 0 | 1 | 2 | 3 | 4 | |
| Baseline (N = 28) | 0 | 14 | 0 | 3 | 13 | 13 | 2 | 0 | 0 | 0.60 (0.62) |
| antisocial first–prosocial second (n = 33) | | | | | | | | | | |
| test trial 1 (after antisocial demonstrator) | 4 | 19 | 8 | 22 | 7 | 6 | 13 | 7 | 0 | 1.61 (1.06) |
| test trial 2 (after prosocial demonstrator) | 3 | 12 | 3 | 14 | 17 | 4 | 8 | 4 | 0 | 0.97 (1.13) |
| prosocial first–antisocial second (n = 28) | | | | | | | | | | |
| test trial 1 (after prosocial demonstrator) | 6 | 16 | 15 | 23 | 4 | 5 | 6 | 9 | 4 | 2.14 (1.30) |
| test trial 2 (after antisocial demonstrator) | 4 | 9 | 6 | 15 | 11 | 6 | 6 | 4 | 1 | 1.21. (1.22) |

prosocial-antisocial/phase 2: z-ratio = 3.955, $p < .001^*$, antisocial-prosocial/phase 1 –antisocial-prosocial/phase 2: z-ratio = 3.166, $p = .006^*$). For a detailed report of stability tests, estimates, standard errors, and confidence intervals see (S1 Text).

## Baseline comparisons

Baseline comparisons revealed that in the *antisocial-prosocial condition*, after the first inefficient demonstration of the antisocial experimenter, children over-imitated above baseline level ($\chi^2 = 19.26$, $df = 1$, $p < .001$, Nagelkerke's $R^2 = .108$), but their over-imitation rates did not exceed baseline level after the efficient demonstration of the prosocial experimenter ($\chi^2 = 3.15$, $df = 1$, $p = .075$, Nagelkerke's $R^2 = .020$). In the *prosocial-antisocial condition*, after the inefficient demonstration of the prosocial experimenter, children's over-imitation rates were above baseline level ($\chi^2 = 38.21$, $df = 1$, $p < .001$, Nagelkerke's $R^2 = .216$), and continued being significantly different from baseline after the efficient demonstration of the antisocial experimenter ($\chi^2 = 7.45$, $df = 1$, $p = .006$, Nagelkerke's $R^2 = .050$). For a detailed report of stability tests, estimates, standard errors, and confidence intervals see (S1 Text).

## Discussion

We conducted an experiment with 5-year-old children who first observed a model retrieve a reward from a puzzle box by performing a series of functionally irrelevant actions and a relevant action. In one condition, the model demonstrating the irrelevant action behaved prosocial in a previous game and the model demonstrating the relevant action behaved antisocial in a previous game. In the other condition these roles were reversed.

According to the dual-process model, we expected children to over-imitate at high rates after the initial inefficient demonstration regardless of whether the actions were demonstrated by the prosocial or antisocial model, but to adjust their imitation rates after the subsequent efficient demonstration. We expected children to stick to the inefficient strategy if it was demonstrated by the prosocial model, more inclined to follow the prosocial model's lead, and at the same time, to contrast their behavior to the antisocial efficient model. In contrast, if they observed the inefficient strategy from an antisocial model and the efficient action from a prosocial model, we expected their over-imitation rates to drop, aligning their behavior with the prosocial efficient strategy model.

These predictions were partly confirmed by our analyses. While we did not find the predicted interaction-effect between condition and test phase, we found the expected pattern of

results in the baseline comparisons. As expected, children showed above-baseline levels of over-imitation, i.e. they copied irrelevant action steps to a considerable degree, regardless of whether the model had acted prosocially or antisocially in a third-party context before engaging in the imitation task. This result confirmed our first hypothesis drawn from the dual-process model of over-imitation [28]: Children likely engaged in Type-1 processing in the first phase of the experiment and blanket copied especially the pseudo-instrumental actions without thoroughly considering their functional relevance or the prior behavior of the model who demonstrated these actions.

In line with previous literature [5, 27, 39], children imitated pseudo-instrumental actions, i.e. actions with direct contact to the reward container, significantly more than no-contact actions. As opposed to no-contact actions, pseudo-instrumental actions are not presumed to trigger a conscious decision to copy or not, but instead induce blanket copying according to the dual-process model of over-imitation. In the second phase of the experiment, all children saw the efficient solution, now demonstrated by the respective other model. In both experimental conditions (prosocial-antisocial and antisocial-prosocial), over-imitation rates dropped from the first to the second test. Importantly, the steepness of this drop did not differ between conditions statistically. This speaks to the notion that children now, after discovering an alternative and more efficient solution to extract a reward, engaged in Type-2 processing: Children were made aware that the previously demonstrated actions have no causal function and questioned whether there are any other reasons to still perform them. We propose that they now actively decided whether to omit or to (continue to) perform the irrelevant actions. As expected, very few children chose to continue over-imitating the antisocial model in the antisocial-prosocial condition. Here, imitation rates went down to baseline level. In the prosocial-antisocial condition, in contrast, some children decided to stick with the prosocial model's inefficient way of extracting the reward and some decided against it and switched to the efficient solution. However, on average over-imitation rates in this condition stayed above baseline level in the second test phase. These baseline comparisons are in line with a dual-process account of over-imitation. While children copy pseudo-instrumental actions (based on Type-1 processing) without questioning the relevancy of the actions or considering the morality of the model in the first phase of the study, they consider these factors in the second phase of the study, once they are made aware that some actions are irrelevant (which activates Type-2 processing). Once Type-2 processes are engaged, contextual cues or individual characteristics influence whether children adapt an instrumental, a normative/conventional, or a ritual stance [20–23, 48].

We need to highlight, that only the comparison with the baseline condition revealed a differential effect and the expected pattern of results. Other than predicted, the interaction between condition and phase was not significant in our main analysis. The difference between the over-imitation scores between conditions in the second test phase was not big enough to yield a significant interaction effect. Why didn't we find the expected interaction between condition and study phase? One explanation for the smaller-than-expected difference between conditions in the second test phase could be the absence of the experimenters during the over-imitation test. While some children still over-imitated after the experimenters left the room, other children omitted the irrelevant actions in the second test phase of the prosocial-antisocial condition. These children might have been more motivated to continue copying the irrelevant actions demonstrated by the prosocial model with increased social incentivization, e.g. with the experimenters present during the test phase. Given that social pressure was minimized (i.e., children were left alone and informed that they could extract the reward however they wished) and the task had an instrumental goal, it is perhaps not surprising that some of the children decided to omit the functionally irrelevant actions altogether, even in the prosocial-

antisocial condition. Another potential reason could be that a model's previous prosocial or antisocial behavior plays only a minor role for a subsequent demonstrated unrelated actions. In a similar study in which the model's pro- or antisociality was manipulated independent from a subsequent imitation task also only minor effects of that manipulation on children's imitation tendencies were found [37]. It is possible that the effects of a model's morality on children's over-imitation (when Type-2 processes are engaged) would have been increased, if the demonstrated actions had been more ritualistic or repetitive triggering a ritual stance [23, 26]. Considering rituals' function in identity construction [49], children might be especially motivated to copy ritualistic actions, when wanting to identify with another person (such as a prosocial model).

Nevertheless, enough children were motivated to continue aligning their behavior with someone they had observed help another person to keep the over-imitation rates above base-line level, even after the actions this person had shown them were unambiguously identified as unnecessary to attain the goal. We assume that these children have stronger social motivations than the others or might be especially sensitive to moral transgressions and rule violations. They might have liked the prosocial model, and disliked the antisocial model, and therefore continued to over-imitate the prosocial model in order to be more like them. Such preferences could be influenced by interindividual differences whose impact is most likely enhanced once Type-2 processes are active: Personality differences, for example in children's sociability, or children's experienced parenting styles, for example whether parents highlighted interdependent vs. independent socialization goals might play a role.

It is also conceivably that children considered the prosocial model to be a better teacher of social norms than the antisocial model, a proven moral transgressor. Similar to an earlier study on food preferences in infants [35], these two aspects are difficult to disentangle in our study and both may even be closely interlinked. In fact, social norms and rituals have been shown to strengthen children's in-group affiliation [50]. Although children differentiate between moral norms and arbitrary social norms such as game rules by three years of age, they would expect in-group members such as the models in the current experiments to adhere to both kinds of norms [38]. Thus, children may well have inferred that a moral transgressor would be more likely to not know or care about the "proper" way to extract the reward. According to the dual-process model of over-imitation, both a wish to affiliate with the model and a motivation to learn and follow norms would lead to over-imitation when Type-2 processes are engaged.

Another aspect that should be considered is that children may have identified the antisocial model as a bully [51]. A bully exerts fear-based power and may force others to adhere to their rules through (threat of) violence. If children in the antisocial-prosocial condition interpreted the antisocial model's third-party moral transgression as bullying, they might have copied the model's actions out of fear. However, we deem this interpretation unlikely. Children were always left alone during the imitation phase in order to reduce such potentially experienced social pressures and even infants do not expect others to continue adhering to a bully's rules once the bully has left the scene (as opposed to their expectations for respect-based power, [51]). Consistent with previous studies using the same task, but manipulating different aspects of the context [16, 39], children showed significantly reduced over-imitation following presentation of an efficient solution in the second phase of the experiment by a second model when both models were equally communicative. Interestingly, differing from our morality manipulation in the current study, an arbitrary minimal group allocation of the child to the inefficient vs. efficient model did not affect their retention of non-functional actions in the second phase of the experiment [16]. At least for the current task this indicates that the models' (a)moral behavior in a previous encounter may have a stronger effect than an arbitrary minimal group

allocation on children's decision whether to retain a socially learned inefficient action strategy or not.

The current study has some limitations. We did not perform a manipulation check on the moral transgression manipulation. Given that the same paradigm has been used extensively in the developmental literature [29, 31, 38], we are confident that children registered both the antisocial and prosocial actions of the respective models and were thus influenced accordingly in their imitation behavior in the second phase of the over-imitation task. Furthermore, to be able to explain the smaller-than-expected difference in over-imitation scores after the efficient action demonstration, it would have been interesting to look into individual differences: Do children who identify more with the prosocial model, children who have stronger social motivations, or children who have a stronger sense of moral responsibility have a higher probability to continuously over-imitate this model than children who do not identify with prosocial individuals as much? It should also be noted that this study was conducted in a western, educated, industrialized, rich, and democratic (WEIRD) population [52]. It is conceivable that the prosocial/antisocial manipulation would have led to stronger effects in cultures that put greater emphasis on relational rather than autonomous socialization goals [53, 54].

To conclude, we show that preschool-aged children initially over-imitate an antisocial model, speaking to a tendency to engage in blanket copying without deliberate consideration of what actions to copy and from whom. Once children encounter the efficient solution to the task, however, social motivations start to play a role and children are more likely to maintain above-baseline level over-imitation of irrelevant actions that were demonstrated by a prosocial model than an antisocial model. Together, the results provide support for the dual-process model of over-imitation [28].

## Supporting information

**S1 Text. Detailed description statistical analysis and results.**
(DOCX)

## Acknowledgments

We thank all the kindergartens that supported our work and all the children who participated in the study.

## Author Contributions

**Conceptualization:** Stefanie Hoehl.

**Data curation:** Hanna Schleihauf.

**Formal analysis:** Hanna Schleihauf.

**Funding acquisition:** Stefanie Hoehl.

**Investigation:** Hanna Schleihauf.

**Methodology:** Hanna Schleihauf.

**Project administration:** Hanna Schleihauf.

**Resources:** Stefanie Hoehl.

**Supervision:** Stefanie Hoehl.

**Visualization:** Hanna Schleihauf.

**Writing – original draft:** Hanna Schleihauf, Stefanie Hoehl.

**Writing – review & editing:** Hanna Schleihauf, Stefanie Hoehl.

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
