## [Decision Letter · Decision Letter 0]

3 Jun 2021

PONE-D-21-14938

Children imitate irrelevant actions performed by the perpetrator of a third-party moral transgression

PLOS ONE

Dear Dr. Schleihauf,

Thank you for submitting your manuscript to PLoS ONE. It has now been read by two experts in this area of research. As you can see in their reviews below, both were positively disposed towards your work. Each review is clear and detailed so I will not elaborate on them further here, but each make excellent points that deserve to be addressed – and I concur with R2 that a video demonstrating the method would aid reader comprehension and facilitate reproduction/extension of your work. Deletion of reference to submitted work is similarly sound advice. I am confident that you can achieve the called-for changes in a re-submission and, in the spirit of transparency, do not intend sending any revised paper out for a further round of review.

We look forward to receiving your revised manuscript.

Kind regards,

Mark Nielsen, Ph.D.

Academic Editor

PLOS ONE

Journal Requirements:

Reviewers' comments:

Reviewer's Responses to Questions

**Comments to the Author**

1. Is the manuscript technically sound, and do the data support the conclusions?

Reviewer #1: Yes

Reviewer #2: Yes

2. Has the statistical analysis been performed appropriately and rigorously? 

Reviewer #1: Yes

Reviewer #2: Yes

3. Have the authors made all data underlying the findings in their manuscript fully available?

Reviewer #1: Yes

Reviewer #2: Yes

4. Is the manuscript presented in an intelligible fashion and written in standard English?

Reviewer #1: Yes

Reviewer #2: Yes

5. Review Comments to the Author

Reviewer #1: This paper presents a study testing the 'dual-process account' of overimitation. Preschool-aged children first watched either a 'helper' or a 'perpetrator' demonstrating an inefficient approach to complete a goal-oriented task. In this phase, children displayed equivalent imitation rates regardless of the nature of the model. In a subsequent phase, they were shown a more efficient approach on the same task by either the same 'perpetrator' or 'helper' model (an alternate model, depending on who they had seen in phase 1). Imitation rates dropped for all children, but they were more likely to still include some redundant actions when the inefficient approach was introduced by the 'helper' (due to stronger social motivation). Children seemed to have employed Type-1 process in phase 1 and Type-2 process in phase 2.

The design is neat, and data analyses are sound. I think the manuscript is well written. The authors seem to know the literature/area of overimitation well. The ideas, findings and results interpretation are clear. Please see below for some specific suggestions/queries for the authors' consideration:

1. I am not certain that the current title is reflective of the key 'take home message' of this study. My impression is that it only covers the first part of the key findings but not the other. The authors might want to consider rephrasing it to more accurately reflect the important findings (I think the fourth 'research highlights' have captured that pretty well).

2. The Introduction covered key theories and previous studies relevant to the current study design. However, they did not include the 'ritual stance' which I think is relevant to the study (see works by Nielsen, Kapitany, or Legare), specifically regarding the discussion/description of no-contact actions. For example, the necklace task used in both studies of Wilks et al (cited across P. 7 - 8) included ritual-like actions. My understanding of the 'ritual stance' is that when certain actions are causally opaque and repetitive, it serves as a strong cue to drive faithful imitation. Would this be considered a kind of Type-1 process? My impression of the no-actions included in this study is that they are not 'ritual-like', but I think it is worth clarifying this as the 'ritual stance' seems to be an emerging account for overimitation. It may also be worth discussing this point in the Discussion.

3. My understanding is that the current apparatus was used previously by the same authors, using similar methodological design(s) with different social factors, though I am not 100% sure if the irrelevant actions in this study were the same or not. It would be interesting to discuss if the current pattern of results was consistent with those found previously (when the social context was set up differently, e.g., in-group vs out-group).

4. The authors raised some interesting suggestions regarding individual difference measures, e.g., social motivation. I was wondering if parenting styles might also play a role in influencing children's imitative tendency (e.g., authoritarianism, interdependent vs independent socialization goals)?

5. A minor comment - the authors used 'pro-social vs anti-social', 'likeable vs dislikable', 'helper vs perpetrator' inter-changeable throughout the manuscript. I appreciate how this helps readers to visualise the manipulations/characteristics of models easier, however, it might be beneficial (less confusing) to keep them consistent (but still keeping those descriptives in the Introduction or Discussion).

Reviewer #2: This is a well-conceived study that challenges young childrens’ striking tendency for over-imitation by including a model who performs a really quite strongly anti-social act, ripping up and abusing someone’s drawing efforts, compared with a model who does not act in the same way. That children still over-imitated in both conditions is testament to their default disposition to blanket copy. The other experimental conditions, notably the second test when the models swap to demonstrate an efficient alternative to the inefficient actions used in the first test, but also the inclusion of the contact versus non-contact actions, all generated interesting results. All these elements and the results they demonstrate will make a significant contribution to the field.

I really have no major nor even many minor critical comments to make. I find the highlights and indeed the whole paper intelligently and clearly written and I have no quibbles with the methods, statistical work, referencing or fair discussion. I do have one suggestion for improvement with respect to the methods reporting – could the authors supply some video illustration of the conduct of the experiment, for supplementary information? The description on page 13 is good as far as it goes, but just exactly how interactions with a child are handled can much affect the results, so to facilitate any later replications by others I think a video illustration would help a lot. Perhaps that will present problems in permissions, but it takes only one parent and child permission to underwrite this. No doubt the action editor can advise further on this.

On line 496, I am not sure what evidence could be confirmed to show a child “deliberately considered” what to do – what are the criteria for “deliberately”? Could that not just say “decided”.

On line 487 the authors cite a ms as submitted – does the journal allow that? Some do not and personally I think they should not, allowing citing of papers in print or in press only/

Apart for that I picked up only a small set of typos for minor correction.

Figure 2 – button not botton –

By line:

269 – quite – quiet

330 – Hewlett –

425 – just say “see supplementary material”

427 – not imitated

END

6. PLOS authors have the option to publish the peer review history of their article (what does this mean?). If published, this will include your full peer review and any attached files.

Reviewer #1: No

Reviewer #2: No

---

## [Editor Report · Decision Letter 1]

11 Aug 2021

Evidence for a dual-process account of over-imitation: Children imitate  anti- and prosocial models equally, but prefer prosocial models once they become aware of multiple solutions to a task

PONE-D-21-14938R1

Dear Dr. Schleihauf,

Thank you for taking the time to revise your paper. I am satisfied with the responses made to the reviewer feedback and the associated changes in the manuscript. We’re thus pleased to inform you that your manuscript has been judged scientifically suitable for publication and will be formally accepted for publication once it meets all outstanding technical requirements.

Kind regards,

Mark Nielsen, Ph.D.

Academic Editor

PLOS ONE

---

## [Editor Report · Acceptance letter]

27 Aug 2021

PONE-D-21-14938R1 

Evidence for a dual-process account of over-imitation: Children imitate anti- and prosocial models equally, but prefer prosocial models once they become aware of multiple solutions to a task 

Dear Dr. Schleihauf:

I'm pleased to inform you that your manuscript has been deemed suitable for publication in PLOS ONE. Congratulations! Your manuscript is now with our production department. 

Kind regards, 

on behalf of

Dr. Mark Nielsen 

Academic Editor

PLOS ONE